# Oncologic outcomes after resection of para-aortic lymph node metastasis in left-sided colon and rectal cancer

**Junichi Sakamoto**[ID]*, **Heita Ozawa**[©], **Hiroki Nakanishi**[©], **Shin Fujita**[©]

Department of Colorectal Surgery, Tochigi Cancer Center, Utsunomiya, Tochigi, Japan

© These authors contributed equally to this work.
* jsakamoto@tochigi-cc.jp

## Abstract

### Aim

The optimal surgical management strategy for para-aortic lymph node (PALN) metastasis has not attracted as much attention as surgery for liver or lung metastasis. The purpose of this retrospective study was to evaluate the oncologic outcomes after synchronous resection of PALN metastasis in left-sided colon and rectal cancer.

### Methods

Between January 1986 and August 2016, 29 patients with pathologically positive PALN metastases who underwent curative resection at our hospital were retrospectively reviewed. We examined clinicopathological characteristics, long-term oncologic outcomes, and factors related to favorable prognosis in these patients.

### Results

The 3-year overall survival and recurrence-free survival (RFS) rates were 50.5% and 17.2%, respectively. In total, 6 (20.7%) patients experienced no recurrence in the 3 years after surgery, while postoperative complications were seen in 9 (31.0%) patients. The 3-year RFS rate was significantly better in the pM1a group than in the pM1b/pM1c group (26.3% and 0.0%, respectively, p = 0.032).

### Conclusion

PALN dissection for patients without other organ metastases in left-sided colon or rectal cancer is a good indication as it is for liver and lung metastasis.

**Data Availability Statement:** All relevant data are available from FigShare (https://doi.org/10.6084/m9.figshare.13168589.v1).

**Funding:** The authors received no specific funding for this work.

**Competing interests:** The authors have declared that no competing interests exist.

## Introduction

Para-aortic lymph node (PALN) metastasis occurs in less than 1.3% of colorectal cancer (CRC) patients [1] and is associated with a poor prognosis [2].

While the management of metastatic CRC has long been based on systemic chemotherapy, several studies have suggested that more aggressive surgical resection is a potentially curative treatment for liver and lung metastasis in selected patients with acceptable postoperative morbidity [3–5]. Recently, surgical resection has been established as the standard therapy for liver and lung metastases.

However, the effectiveness of surgical management of synchronous PALN metastasis remains highly controversial because of a lack of definitive evidence regarding survival outcomes and the safety of surgical techniques [6]. There is insufficient data to guide the stratification of patients for aggressive treatment.

We aimed to clarify the oncologic outcomes after synchronous resection of PALN metastasis in left-sided colon and rectal cancer.

## Materials and methods

### Ethics statement

This submission does not require an ethics statement. The study protocol was conducted in accordance with the Declaration of Helsinki. All data were fully anonymized before we accessed. The datasets analyzed during the current study are available from the corresponding author on reasonable request. All relevant data are within the paper and its Supporting Information files. The need for written consent from the study subjects was waived by the institutional review board, and this retrospective study was approved by the Ethical Advisory Committee of the Tochigi Cancer Center before study initiation.

### Patients

In this retrospective cohort study, 574 patients with stage Ⅱ CRC underwent surgery, including noncurative surgery, at our cancer center between January 1986 and August 2016. Of these, 43 underwent PALN dissection synchronously with a primary CRC resection.

The selection criteria for a PALN dissection were as follows: (1) pathological diagnosis of CRC; (2) suspected PALN metastasis on preoperative imaging, such as abdominal/pelvic computed tomography (CT) or positron emission tomography; and (3) an assessment that curative resection was possible (i.e., no signs of upward PALN swelling extending above the renal vessels, or an obvious invasion of PALN metastases to the great vessels). Curative resection was defined as complete tumor resection with all margins being negative. The indications for PALN dissection were thoroughly discussed and determined at our multidisciplinary team conferences with radiologists and hepatobiliary surgeons.

In total, 29 patients who were pathologically positive for PALN metastasis were included. We excluded patients with secondary malignancies and double cancer.

### Evaluation parameters

The classification system of the Union for International Cancer Control (8th edition) was used to determine pathological tumor depth and distant metastasis. The extent of regional lymph node metastasis was classified into 3 categories according to their location: (1) pericolic/perirectal lymph nodes were defined as lymph nodes close to the bowel wall; (2) intermediate lymph nodes were defined as lymph nodes along the feeding arteries; and (3) main lymph nodes were defined as lymph nodes related to the origin of the feeding artery. In addition,

lateral pelvic nodes were defined as lymph nodes along the common internal and external iliac vessels, and proceeding downwards to the level of the obturator internus muscles. Postoperative complications were categorized according to the Clavien-Dindo classification.

## Surgery and follow-up

Curative surgery was performed as per the standard procedure of total mesorectal excision. After identification of the ureter and gonadal vessels, PALN dissection commenced from the aorta or bifurcation of the iliac artery. We removed all lymphovascular tissues in the area using the following boundaries: the lower border of the left renal vein, the right border of the inferior vena cava, and the right border of the left gonadal vessels (Fig 1). In the present study, we excluded patients who underwent PA lymphadenectomy.

Patients underwent a standardized follow-up every 3 months for the first 3 years, and at each follow-up, a physical examination and laboratory tests were performed. In addition, CT was performed every 6 months and a colonoscopy was performed 1 year after surgery and repeated at least every 2 years.

## Main outcome measures

The primary end points were 3-year overall survival (OS) and recurrence-free survival (RFS) rates.

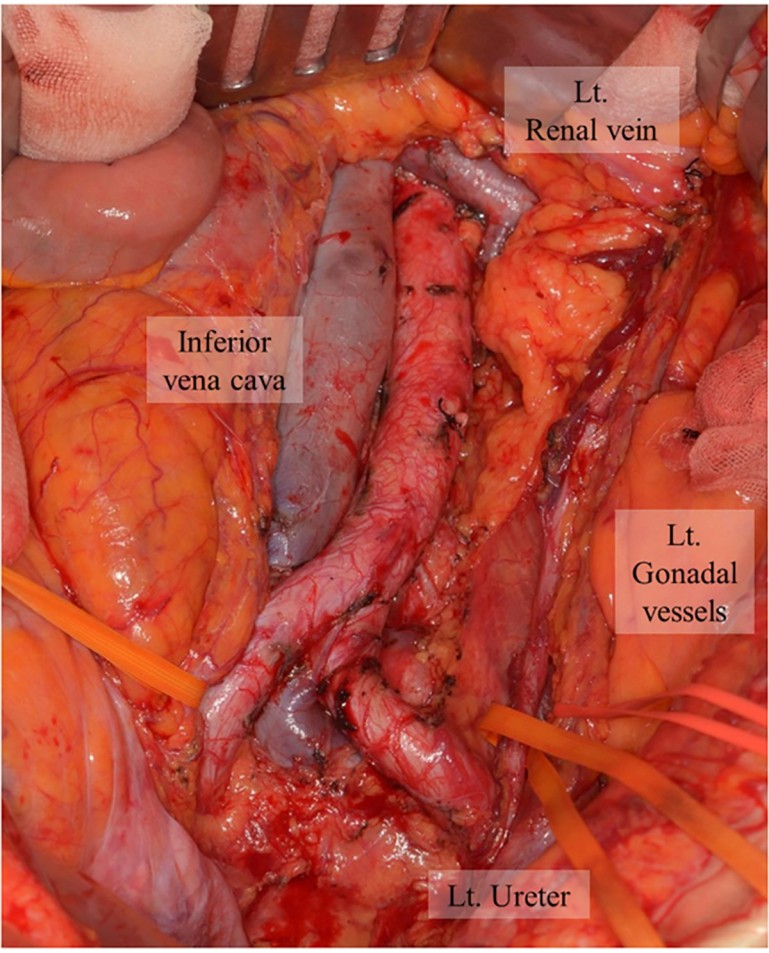

**Fig 1. Intraoperative findings of PALN dissection.** All lymphovascular structures were removed from the lower border of the left renal vein, the right border of the inferior vena cava, and the right border of the left gonadal vessels. PALN, para-aortic lymph node.

## Statistical analysis

All statistical analyses were performed with EZR (Saitama Medical Center, Jichi Medical University, Saitama, Japan), which is a graphical user interface for R (The R Foundation for Statistical Computing, Vienna, Austria). More precisely, it is a modified version of the R Commander designed to add statistical functions that are frequently used in biostatistics.

Differences in categorical and continuous variables were analyzed using the chi-square test (or Fisher's exact test) and Student's *t*-test, while the Kaplan-Meier method was used to compare OS and RFS rates. Univariate and multivariate analyses of factors associated with oncologic outcomes were evaluated using the Kaplan-Meier method, and the Cox proportional hazards model was used to estimate hazard ratios. Survival curves were created using the log-rank test. From the receiver operating characteristic (ROC) curves, the threshold of PALN metastasis was set to 4 (area under the ROC curve, 0.741; 95% CI, 0.531–0.951). A probability level of $p < 0.05$ was considered to indicate statistical significance.

## Results

Clinicopathological characteristics are shown in Table 1. The median age of the patients was 60 years (range: 35–74 years), and 15 (51.7%) were men. In 15 (51.7%) patients, the tumors were located in the rectum. In total, 18 (62.1%) patients received adjuvant treatment. The most common histological type was moderately differentiated adenocarcinoma (n = 19, 65.5%), while 5 (17.2%) patients had pT4b tumors, and 14 (48.3%) had no metastases to the main lymph nodes. In terms of other organ metastases, 19 (65.5%) patients were in the pM1a group. All patients received simultaneous resection of their distant metastases. The median number of total harvested and metastatic PALNs was 12 (1–81) and 4 (1–71), respectively.

Operative and postoperative results are shown in Table 2. Only 1 (3.4%) patient was operated on using a laparoscopic approach. The median operating time was 248 (110–645) minutes, the median estimated blood loss was 628 (20–4900) g, and the median hospital stay was 40 (8–106) days. Postoperative morbidity occurred in 9 (31.0%) patients. There was no 30-day mortality, and no patient had grade Ⅳ or Ⅴ complications. The most common morbidity was surgical site infection (n = 3, 10.3%). Postoperative recurrences occurred in 23 (79.3%) patients, and the most common site of recurrence was the distant lymph nodes (n = 9, 31%).

The median follow-up was 30.0 months (range: 1.5–210.7 months). Of the total 29 patients, the 3-year OS rate was 50.5% (Fig 2A), and the 3-year RFS rate was 17.2% (Fig 2B). Furthermore, the 3-year OS rate in the pM1a group was significantly better than in the pM1b and pM1c groups (63.2 and 24.0%, respectively; hazard ratio [HR], 3.01; 95% confidence interval [CI], 1.19–7.65; p = 0.015) (Fig 3A). In addition, the 3-year RFS rate was significantly different in the pM1a group and the pM1b and pM1c group (26.3 and 0.0%, respectively; HR, 2.49; 95% CI, 1.05–5.90; p = 0.032) (Fig 3B). There were no statistically significant differences in clinicopathological characteristics between patients with pM1a and pM1b and pM1c except for the rate of adjuvant treatment (Table 3).

In multivariate analysis (Table 4), the pM1a group was an independent prognostic factor for OS (HR, 5.15; 95% CI, 1.52–17.5; p = 0.0084) and RFS (HR, 2.49; 95% CI, 1.05–5.90; p = 0.038). The number of PALN metastases did not differ significantly based on the OS or RFS.

## Discussion

The present study demonstrated that PALN dissection for left-sided colon and rectal cancer with synchronous PALN metastasis without other organ metastases was associated with a

**Table 1. Clinicopathological characteristics.**

| n = 29 | | |
|---|---|---|
| Age, years† | 60 | (35–74) |
| Sex, n (%) | | |
| Male | 15 | (51.7) |
| Female | 14 | (48.3) |
| Location of tumor, n (%) | | |
| Left-sided colon | 14 | (48.3) |
| Rectum | 15 | (51.7) |
| Adjuvant treatment, n (%) | | |
| None | 11 | (37.9) |
| Neoadjuvant chemotherapy | 1 | (3.4) |
| Adjuvant chemotherapy | 15 | (51.7) |
| Postoperative chemoradiation therapy | 2 | (6.9) |
| Histology, n (%) | | |
| Well-differentiated adenocarcinoma | 2 | (6.9) |
| Moderately differentiated adenocarcinoma | 19 | (65.5) |
| Poorly differentiated adenocarcinoma | 6 | (20.7) |
| Mucinous adenocarcinoma | 2 | (6.9) |
| Depth of invasion, n (%) | | |
| pT3 | 13 | (44.8) |
| pT4a | 11 | (37.9) |
| pT4b | 5 | (17.2) |
| Extent of lymph node metastasis, n (%) | | |
| Pericolic/perirectal | 23 | (79.3) |
| Intermediate | 19 | (65.5) |
| Main | 15 | (51.7) |
| Lateral pelvic node | 6 | (20.7) |
| Distant metastasis, n (%) | | |
| pM1a (only PALN) | 19 | (65.5) |
| pM1b (PALN and liver metastases) | 7 | (24.1) |
| pM1c (PALN and liver, peritoneal metastases) | 3 | (10.3) |
| Number of harvested PALNs† | 12 | (1–81) |
| Number of metastatic PALNs† | 4 | (1–71) |
| Number of metastatic PALNs, n (%) | | |
| 1–3 | 14 | (48.3) |
| ≧4 | 15 | (51.7) |

†Data are presented as median (range), unless otherwise stated.

PALN, para-aortic lymph node.

favorable prognosis. This is one of a few characteristic studies that has shown the clinical significance of PALN dissection.

Similar to liver and lung metastasis, synchronous PALN metastasis from a CRC is categorized as Stage IV disease. Liver and lung metastasis resections are widely accepted as standard treatments, and the 5-year survival rates are over 50% following surgery [7].

PALN dissection was first described in 1950 by Dr. Deddish as a modification of the Miles abdominoperineal resection performed to reduce local recurrence in rectal cancer [8]. However, routine PALN dissection has since been abandoned in view of increased surgical

**Table 2. Operative and postoperative results.**

| | n = 29 | |
|---|---|---|
| Operative procedure, n (%) | | |
| Open | 28 | (96.6) |
| Laparoscopic | 1 | (3.4) |
| Operation time, min† | 248 | (110–645) |
| Blood loss, g† | 628 | (20–4900) |
| Hospital stay, days† | 40 | (8–106) |
| Morbidity, n (%) | 9 | (31.0) |
| GradeⅠ | | |
| Urinary retention | 1 | (3.4) |
| GradeⅡ | | |
| SSI | 2 | (6.9) |
| Intra-abdominal abcess | 1 | (3.4) |
| Atelectasis | 1 | (3.4) |
| Delayed gastric emptying | 1 | (3.4) |
| GradeⅢ | | |
| SSI | 1 | (3.4) |
| Paralytic ileus | 1 | (3.4) |
| Anastomotic leakage | 1 | (3.4) |
| ≧GradeⅣ | None | |
| Recurrence, n (%) | 23 | (79.3) |
| Distant lymph node | 9 | (31.0) |
| Liver | 8 | (27.6) |
| Peritoneum | 4 | (13.8) |
| Lung | 3 | (10.3) |
| Local recurrence | 2 | (6.9) |
| Bone | 1 | (3.4) |
| Others | 2 | (6.9) |

†Data are presented as median (range), unless otherwise stated.

SSI, surgical site infection.

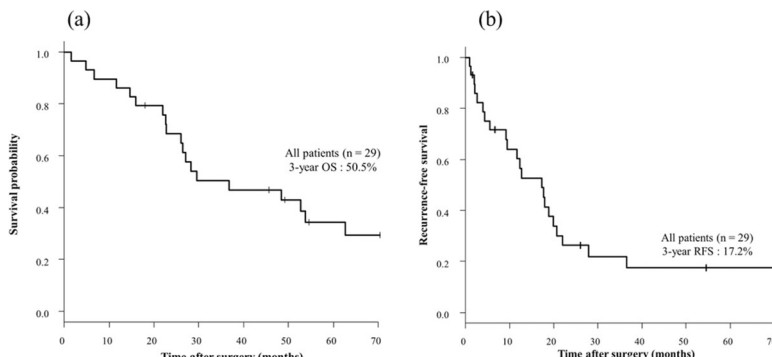

**Fig 2.** Kaplan-Meier overall survival (a) and recurrence-free survival (b) curve for all patients. The 3-year OS rate was 50.5% (Fig 2A), and the 3-year RFS rate was 17.2% (Fig 2B). OS, overall survival; RFS, recurrence-free survival.

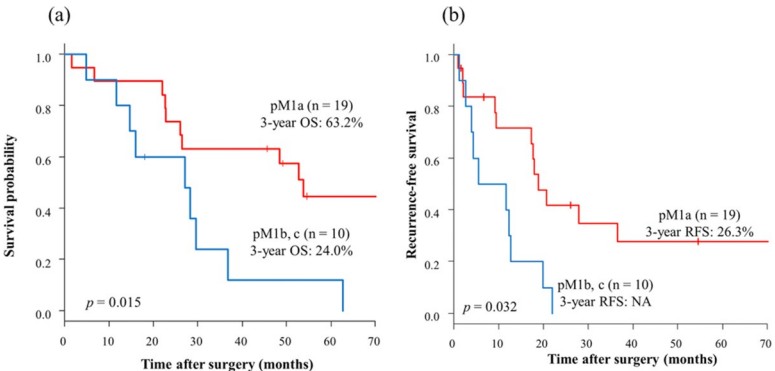

**Fig 3.** Kaplan-Meier overall survival (a) and recurrence-free survival (b) curve for patients with each M-category pM1a and pM1b, c. The 3-year OS rate in the pM1a group was significantly better than that in the pM1b and pM1c groups (63.2 and 24.0%, respectively) (Fig 3A). In addition, the 3-year RFS rate was significantly different in the pM1a group and the pM1b and pM1c groups (26.3 and 0.0%, respectively) (Fig 3B). red, pM1a; blue, pM1b and pM1c. OS, overall survival; RFS, recurrence-free survival; NA, not applicable.

morbidity, such as urinary and sexual dysfunction, without corresponding improvements in recurrence rates and overall survival [9]. On the other hand, recent studies have reported that prolonged survival can be obtained following resection of PALN metastasis [10–14]. Each of these studies was a retrospective cohort study, and so the significance of PALN dissection remains unconfirmed and highly controversial.

We think that curative resection, using PALN dissection, is a prerequisite for a favorable prognosis. Past studies have shown that low curative resection rates lead to low survival rates [12]. Therefore, we did not perform PALN dissection on patients for whom it was established that curative resection was not possible based on preoperative imaging diagnosis.

Our PALN dissection area was similar to that reported in past studies. It is necessary to perform PALN dissections for right-sided colon cancer while maintaining the great vessels, such as the superior mesenteric artery or celiac artery. Because of this, in right-sided colon cancer, systematic PALN dissection is anatomically impossible, and the dissection effect is not attained as it is with left-sided colon and rectal cancers. For this reason, we limited the indication for PALN dissection to left-sided colon and rectal cancers.

The 3-year OS and 3-year RFS rates were significantly better in the pM1a group than in the pM1b/pM1c group, which was similar to the results of Yamada et al [11]. In our study, there were no significant differences in clinicopathological characteristics, such as tumor location and histology, the number of metastatic regional lymph nodes, and the number of metastatic PALNs in the pM1a group and pM1b/pM1c group (Table 3). These results suggest that other organ metastases without PALN are the most important prognosticators. In our study, there was no distant lymph node recurrence in the pM1b/pM1c group, and all recurrences occurred in other organs. Consequently, patients with PALN metastasis with other organ metastases were possibly excluded from the indication for PALN dissection.

Song et al. reported that patients without disease recurrence had 3 or fewer PALN metastases [14]. Several other studies reported that fewer metastases may be a good indication for PALN dissection [2, 15]. With regards to f lateral lymph nodes in lower rectal cancer, Fujita reported that the prognosis of patients with 1 or 2 extramesorectal lymph node metastases was favorable [16]. Our data, however, showed that 3 patients achieved long-term RFS, even when the number of PALN metastases reached 7 or more. Additionally, there are very few reports that detail the relationship between the number of metastatic PALNs and prognosis, so no

**Table 3. Comparison of clinicopathological characteristics between the pM1a and pM1b/c.**

| Variable | pM1a (n = 19) | pM1b and pM1c (n = 10) | p value |
|---|---|---|---|
| Age, years† | 63 (46–74) | 59.5 (35–74) | 0.25 |
| Sex, n (%) | | | 0.25 |
| Male | 8 (42.1%) | 7 (70.0%) | |
| Female | 11 (57.9%) | 3 (30.0%) | |
| Location of tumor, n (%) | | | 1 |
| Left-sided colon | 9 (47.3%) | 4 (40.0%) | |
| Rectum | 10 (52.6%) | 6 (60.0%) | |
| Histology, n (%) | | | 0.68 |
| Well or Moderately | 13 (68.4%) | 8 (80.0%) | |
| Poorly or Mucinous | 6 (31.6%) | 2 (20.0%) | |
| Depth of invasion, n (%) | | | 1 |
| pT3 | 9 (47.3%) | 4 (40.0%) | |
| pT4a or pT4b | 10 (52.6%) | 6 (50.0%) | |
| Number of harvested regional LNs† | 25 (17–155) | 51 (17–165) | 0.49 |
| Number of metastatic regional LNs† | 7 (1–37) | 19.5 (4–122) | 0.10 |
| Number of harvested PALNs† | 11 (1–45) | 14.5 (3–81) | 0.29 |
| Number of metastatic PALNs† | 2 (1–25) | 5.5 (1–71) | 0.17 |
| Adjuvant treatment, n (%) | | | 0.044 |
| Yes | 9 (47.3%) | 9 (90.0%) | |
| No | 10 (52.6%) | 1 (10.0%) | |

†Data are presented as median (range), unless otherwise stated.

LN, lymph node

PALN, para-aortic lymph node

**Table 4. Univariate and multivariate analyses of overall survival and recurrence-free survival.**

| Variable | | Overall survival | | | | | Recurrence-free survival | | | | |
|---|---|---|---|---|---|---|---|---|---|---|---|
| | Number | Univariate | | | Multivariate | | Univariate | | | Multivariate | |
| | | 3-year OS [%] | HR [95% CI] | p value | HR [95% CI] | p value | 3-year RFS [%] | HR [95% CI] | p value | HR [95% CI] | p value |
| Tumor location | | | | | | | | | | | |
| Colon | 13 | 46.2 | 1 | 0.72 | | | 23.1 | 1 | 0.86 | | |
| Rectum | 16 | 53.6 | 0.84 [0.35–2.05] | | | | 12.5 | 0.48 [0.19–1.21] | | | |
| Histology | | | | | | | | | | | |
| Well or Moderately | 21 | 65.3 | 1 | 0.0010 | 1 | 0.0011 | 19.0 | 1 | 0.42 | | |
| Poorly or Mucinous | 8 | 12.5 | 4.21 [1.67–10.6] | | 7.18 [2.21–23.4] | | 12.5 | 2.20 [0.81–5.99] | | | |
| Distant metastasis | | | | | | | | | | | |
| M1a | 19 | 63.2 | 1 | 0.015 | 1 | 0.0084 | 26.3 | 1 | 0.032 | 1 | 0.038 |
| M1b and M1c | 10 | 24.0 | 3.01 [1.19–7.65] | | 5.15 [1.52–17.5] | | 0 | 3.59 [1.15–11.21] | | 2.49 [1.05–5.90] | |
| Number of PALN metastases | | | | | | | | | | | |
| 1–3 | 14 | 62.3 | 1 | 0.029 | 1 | 0.79 | 28.6 | 1 | 0.11 | | |
| 4 and more | 15 | 40.0 | 2.81 [1.07–7.39] | | 1.16 [0.38–3.56] | | 6.7 | 2.15 [0.68–6.82] | | | |
| Adjuvant treatment | | | | | | | | | | | |
| Yes | 18 | 36.1 | 1.91 [0.73–5.00] | 0.18 | | | 22.2 | 1 | 0.57 | | |
| No | 11 | 72.7 | 1 | | | | 9.1 | 0.35 [0.13–0.96] | | | |

HR, hazard ratio; CI, confidence interval; OS, overall survival; RFS, recurrence-free survival; PALN, para-aortic lymph node.

influencing factors have been established. Consequently, the number of metastatic PALNs cannot guide the indication for PALN dissection at this moment.

Recently, several studies have reported an optimum size cutoff for lymph nodes to identify patients positive for lateral lymph node metastases of lower rectal cancer on preoperative imaging [17, 18]. However, reports on preoperative diagnosis of PALN metastasis are rare. Further studies on accurate preoperative imaging diagnosis and patient selection for PALN dissection are therefore necessary.

The benefits associated with removing PALN metastasis should be weighed up against the risk of morbidity. In the present study, postoperative morbidity occurred in 31.0% of patients, which was comparable with that of other studies (7.8–38.9%) [10–14]. The main morbidity was surgical site infection, and the rate of Clavien and Dindo classification grade III or above was only 10.3%, with no perioperative deaths. These results suggest that the incidence of postoperative morbidities associated with PALN dissection is within acceptable limits.

There were several limitations to the present study, including the single institutional experience, the small sample size due to the rarity of this metastatic pattern, and the retrospective analysis. The study period was long, lasting over 30 years; and during this time, the optimal indication for PALN dissection and treatment strategy, e.g. chemotherapeutic regimens, would have changed. Additionally, an assessment of sexual dysfunction was not performed. A global assessment method, such as the International Index of Erectile Function, should be used for all cases. Finally, the present study did not include the patients without PALN dissection. Consequently, the efficacy of PALN dissection cannot be predicted with total accuracy based on our results. Larger multi-institutional prospective studies are required to overcome the shortcomings of this research. However, our results clearly showed that a favorable prognosis could be expected in selecting patients with left-sided colon and rectal cancers using isolated PALN dissection.

## Conclusions

PALN dissection for patients without other organ metastases in left-sided colon or rectal cancer is a good indication as it is for liver and lung metastasis.

## Supporting information

**S1 Dataset.**
(PDF)

## Acknowledgments

This study was supported by the Tochigi Cancer Center founded by Tochigi prefecture. All authors listed have contributed sufficiently to this study to be included as authors.

## Author Contributions

**Conceptualization:** Junichi Sakamoto.

**Data curation:** Junichi Sakamoto, Hiroki Nakanishi.

**Formal analysis:** Junichi Sakamoto.

**Funding acquisition:** Junichi Sakamoto.

**Investigation:** Junichi Sakamoto.

**Methodology:** Junichi Sakamoto.

**Project administration:** Junichi Sakamoto.

**Resources:** Junichi Sakamoto.

**Software:** Junichi Sakamoto.

**Supervision:** Junichi Sakamoto.

**Validation:** Junichi Sakamoto.

**Visualization:** Junichi Sakamoto.

**Writing – original draft:** Junichi Sakamoto.

**Writing – review & editing:** Heita Ozawa, Shin Fujita.

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
