## [Decision Letter · Decision Letter 0]

29 Jul 2020

PONE-D-20-18946

Oncologic outcomes after resection of para-aortic lymph node metastasis in left-sided colon and rectal cancer

PLOS ONE

Dear Dr. Sakamoto,

Thank you for submitting your manuscript to PLOS ONE. After careful consideration, we feel that it has merit but does not fully meet PLOS ONE’s publication criteria as it currently stands. Therefore, we invite you to submit a revised version of the manuscript that addresses the points raised during the review process.

We look forward to receiving your revised manuscript.

Kind regards,

Norikatsu Miyoshi, M.D., Ph.D., FACS

Academic Editor

PLOS ONE

Journal Requirements:

2. Our staff editors have determined that your manuscript is likely within the scope of our Cancer Metastasis Call for Papers. This editorial initiative is headed by a team of Guest Editors for PLOS ONE: Joe Ramos (University of Hawai'i), Shengyu Yang (Penn State University), Helen Fillmore (University of Portsmouth) and Tobias Zech (University of Liverpool). The Collection will encompass a diverse range of research articles about metastasis, including mechanisms of cell motility, invasion and the tumor microenvironment, as well as advances in the development of anti-metastatic therapies.  Additional information can be found on our announcement page: https://collections.plos.org/s/cancer-metastasis

If you would like your manuscript to be considered for this collection, please let us know in your cover letter and we will ensure that your paper is treated as if you were responding to this call.  Please note that being considered for the Collection does not require additional peer review beyond the journal’s standard process and will not delay the publication of your manuscript if it is accepted by PLOS ONE. If you would prefer to remove your manuscript from collection consideration, please specify this in the cover letter.

3. In the ethics statement in the manuscript and in the online submission form, please provide additional information about the patient records used in your retrospective study, including: a) whether all data were fully anonymized before you accessed them; b) the date range (month and year) during which patients' medical records were accessed; and d) the source of the medical records analyzed in this work (e.g. hospital, institution or medical center name). If patients provided informed written consent to have data from their medical records used in research, please include this information.

5. Your ethics statement must appear in the Methods section of your manuscript. If your ethics statement is written in any section besides the Methods, please move it to the Methods section and delete it from any other section. Please also ensure that your ethics statement is included in your manuscript, as the ethics section of your online submission will not be published alongside your manuscript.

Reviewers' comments:

Reviewer's Responses to Questions

**Comments to the Author**

1. Is the manuscript technically sound, and do the data support the conclusions?

Reviewer #1: Partly

Reviewer #2: Partly

2. Has the statistical analysis been performed appropriately and rigorously? 

Reviewer #1: Yes

Reviewer #2: Yes

3. Have the authors made all data underlying the findings in their manuscript fully available?

Reviewer #1: Yes

Reviewer #2: Yes

4. Is the manuscript presented in an intelligible fashion and written in standard English?

Reviewer #1: Yes

Reviewer #2: Yes

5. Review Comments to the Author

Reviewer #1: This article was described about survival after para-aortic lymph node dissection of the colorectal cancer patients.

Given the nature of this relatively small study, caution needs to be exercised to interpret the presented findings.

But, the report was well written.

Comment

In page23, line 49-50, the authors described that patients with PALN metastasis with other organ metastasis should be excluded from indication for PALN dissection. It is misunderstanding by these results that all the patients underwent the surgery in this series.

This is not a report of the efficacy of PALN dissection for the patients with pM1b/c compared with those who did not underwent PALND.

Certainly, the survival of the patients with pM1a is superior to that of other patients.

The PALN dissection is good indicated for the patients with pM1a, from this report.

Nevertheless, all recurrence after R0 resection of the patients with pM1b/c occurred distant other organ, which means that the PALND potentially have certain treatment effect of local control in the pM1b/c cohort, especially in the era of chemotherapy matured.

The study period was extremely long, including the period of chemotherapy drug lag. From this paper, anyone cannot deny that the combination of intensive and intensified chemotherapy and PALND may be effective for the patients, even if with pM1b/c and PALN metastasis.

It is appropriate to use the term “maybe excluded” or “possibly excluded from indication for PALN dissection”

It is right to emphasize that “pM1a is good indication” in conclusion.

Minor comments

# Does the study include both patients who underwent PALN dissection or PA lymphadenectomy?

# In page18, line2-5, the authors said that there were no statistically significant differences in clinico-pathological characteristics between patients with pM1a and pM1b and pM1c.

The rate of adjuvant treatment is different between two groups.

Reviewer #2: In this study, authors evaluated the oncologic outcomes after synchronous resection of PALN metastasis in left-sided colon and rectal cancer as a retrospective study. The 3- year RFS rate was significantly better in the pM1a group than in the pM1b/pM1c group, and they concluded PALN dissection for left-sided colon or rectal cancer with synchronous PALN metastasis can be a feasible treatment. The aim of this study and their opinion are understandable. However, there are some points to be revised.

They selected patients with curative surgery. There were 7 M1b patients and 3 M1c patients.　 Please describe the details of these metastases and the surgery (simultaneous surgery or irregular surgery). Also, please indicate from which point you started calculating RFS and OS.

They said that patients with PALN metastasis with other organ metastases should be excluded from the indication for PALN dissection in discussion section. Patients with M1b/M1c had a poor prognosis and may not need PALN resection. However, regarding whether PALN dissection is effective in patients with M1a, it is necessary to compare the prognosis of patients with and without PALN dissection. Is this comparison possible with your dataset?

6. PLOS authors have the option to publish the peer review history of their article (what does this mean?). If published, this will include your full peer review and any attached files.

Reviewer #1: No

Reviewer #2: No

---

## [Author Response · Author response to Decision Letter 0]

11 Sep 2020

August 9, 2020

Norikatsu Miyoshi, M.D., Ph.D., FACS

Academic Editor

PLOS ONE

Re: Manuscript ID:PONE-D-20-18946

Oncologic outcomes after resection of para-aortic lymph node metastasis in left-sided colon and rectal cancer

Dear Dr. Norikatsu Miyoshi:

Thank you for the thoughtful and constructive feedback you provided regarding our manuscript (PONE-D-20-18946), titled “Oncologic outcomes after resection of para-aortic lymph node metastasis in left-sided colon and rectal cancer.” We also appreciate the time and effort you and each of the reviewers have dedicated to providing insightful feedback on ways to strengthen our paper. Thus, it is with great pleasure that we resubmit our article for further consideration. We have incorporated changes that reflect the detailed suggestions you have graciously provided. We also trusted that our edits and the responses we provided below satisfactorily address all the issues and concerns you and the reviewers have noted.

To facilitate your review of our revisions, the following is a point-by-point response to the questions and comments delivered in your letter.

Reviewer #1

This article was described about survival after para-aortic lymph node dissection of the colorectal cancer patients. Given the nature of this relatively small study, caution needs to be exercised to interpret the presented findings. But, the report was well written.

(Response) We thank you for providing constructive comments regarding the improvement of the original manuscript.

(Comment 1) 

In page23, line 49-50, the authors described that patients with PALN metastasis with other organ metastasis should be excluded from indication for PALN dissection. It is misunderstanding by these results that all the patients underwent the surgery in this series. This is not a report of the efficacy of PALN dissection for the patients with pM1b/c compared with those who did not underwent PALND. Certainly, the survival of the patients with pM1a is superior to that of other patients. The PALN dissection is good indicated for the patients with pM1a, from this report. Nevertheless, all recurrence after R0 resection of the patients with pM1b/c occurred distant other organ, which means that the PALND potentially have certain treatment effect of local control in the pM1b/c cohort, especially in the era of chemotherapy matured. The study period was extremely long, including the period of chemotherapy drug lag. From this paper, anyone cannot deny that the combination of intensive and intensified chemotherapy and PALND may be effective for the patients, even if with pM1b/c and PALN metastasis. It is appropriate to use the term “maybe excluded” or “possibly excluded from indication for PALN dissection”. It is right to emphasize that “pM1a is good indication” in conclusion. 

(Response) We appreciate the Reviewer’s comment on this point. The current study did not include the patients without PALN dissection. It is inappropriate to use the term “should be excluded”. Accordingly, we have changed the following text in the Discussion (page21, lines 50-51): “Consequently, patients with PALN metastasis with other organ metastases should be excluded from the indication for PALN dissection.” to “Consequently, patients with PALN metastasis with other organ metastases were possibly excluded from the indication for PALN dissection.” Moreover, we have changed the next text in the Abstract (page8, lines19-21) and Conclusion (page23, lines98-100): “PALN dissection for patients without other organ metastases in left-sided colon or rectal cancer can be a feasible treatment option as it is for liver and lung metastasis.” to “PALN dissection for patients without other organ metastases in left-sided colon or rectal cancer is a good indication as it is for liver and lung metastasis.”

(Comment 2)

Does the study include both patients who underwent PALN dissection or PA lymphadenectomy?

 (Response) As the Reviewer noted, our original expression here may be a bit misleading. In this study, we excluded patients who underwent non-systematical PALN dissection. Accordingly, we have added the following text in Materials and methods (page11, lines 22-23): “In the present study, we excluded patients who underwent PA lymphadenectomy.”

(Comment 3)

In page18, line2-5, the authors said that there were no statistically significant differences in clinico-pathological characteristics between patients with pM1a and pM1b and pM1c. The rate of adjuvant treatment is different between two groups. 

(Response) The reviewer’s comment is correct. To clarify, we have revised as “There were no statistically significant differences in clinicopathological characteristics between patients with pM1a and pM1b and pM1c besides the rate of adjuvant treatment.” (page17, lines2-5)

Reviewer #2

(Response) We wish to express our appreciation to the Reviewer for his or her insightful comments, which have helped us significantly improve the paper.

(Comment 1)

In this study, authors evaluated the oncologic outcomes after synchronous resection of PALN metastasis in left-sided colon and rectal cancer as a retrospective study. The 3- year RFS rate was significantly better in thepM1a group than in the pM1b/pM1c group, and they concluded PALN dissection for left-sided colon or rectal cancer with synchronous PALN metastasis can be a feasible treatment. The aim of this study and their opinion are understandable. However, there are some points to be revised.

They selected patients with curative surgery. There were 7 M1b patients and 3 M1c patients.　 Please describe the details of these metastases and the surgery (simultaneous surgery or irregular surgery). Also, please indicate from which point you started calculating RFS and OS.

They said that patients with PALN metastasis with other organ metastases should be excluded from the indication for PALN dissection in discussion section. Patients with M1b/M1c had a poor prognosis and may not need PALN resection. However, regarding whether PALN dissection is effective in patients with M1a, it is necessary to compare the prognosis of patients with and without PALN dissection. Is this comparison possible with your dataset?

 (Response)　In the present study, the site of other organ metastases were the liver metastases in 7 patients (pM1b group), and the liver and peritoneal metastases in 3 patients (pM1c group). All patients received simultaneous resection with their primary cancer. Accordingly, we have revised the Table 1 as follows (page14): 

Moreover, we have added the following text to the Results (page13, lines22-23): “All patients received simultaneous resection of their distant metastases.” Further, as the Reviewer noted, this is not a report of the efficacy of PALN dissection for the patients with pM1a compared with those who did not undergo PALN dissection. This comparison is impossible with our dataset. We have therefore changed and added the following text as one of the limitations of the study (page23, lines88-95): “Finally, the present study did not include the patients without PALN dissection. Consequently, the efficacy of PALN dissection cannot be predicted with total accuracy based on our results. Larger multi-institutional prospective studies are required to overcome the shortcomings of this research. However, our results clearly showed that a favorable prognosis could be expected in selecting patients with left-sided colon and rectal cancers using isolated PALN dissection.” Finally, we have changed the next text in the Abstract (page8, lines19-21) and Conclusion (page23, lines98-100): “PALN dissection for patients without other organ metastases in left-sided colon or rectal cancer can be a feasible treatment option as it is for liver and lung metastasis.” to “PALN dissection for patients without other organ metastases in left-sided colon or rectal cancer is a good indication as it is for liver and lung metastasis.”

With these changes to our manuscript, we hereby resubmit our manuscript for a secondary evaluation. Again, thank you for allowing us to strengthen our manuscript with your valuable comments and queries. We have worked hard to incorporate your feedback and hope that these revisions persuade you to accept our submission.

Sincerely,

Junichi Sakamoto, MD

Department of Colorectal Surgery

Tochigi Cancer Center

4 Chome-9-13 Yonan, Utsunomiya, Tochigi 320-0834, Japan

Tel: 028-658-5151

Fax: 028-658-5669

Email: jsakamoto@tochigi-cc.jp

---

## [Decision Letter · Decision Letter 1]

21 Oct 2020

Oncologic outcomes after resection of para-aortic lymph node metastasis in left-sided colon and rectal cancer

PONE-D-20-18946R1

Dear Dr. Sakamoto,

We’re pleased to inform you that your manuscript has been judged scientifically suitable for publication and will be formally accepted for publication once it meets all outstanding technical requirements.

Kind regards,

Norikatsu Miyoshi, M.D., Ph.D., FACS

Academic Editor

PLOS ONE

Additional Editor Comments (optional):

Reviewers' comments:

Reviewer's Responses to Questions

**Comments to the Author**

1. If the authors have adequately addressed your comments raised in a previous round of review and you feel that this manuscript is now acceptable for publication, you may indicate that here to bypass the “Comments to the Author” section, enter your conflict of interest statement in the “Confidential to Editor” section, and submit your "Accept" recommendation.

Reviewer #1: All comments have been addressed

Reviewer #2: All comments have been addressed

2. Is the manuscript technically sound, and do the data support the conclusions?

Reviewer #1: Partly

Reviewer #2: Yes

3. Has the statistical analysis been performed appropriately and rigorously? 

Reviewer #1: Yes

Reviewer #2: Yes

4. Have the authors made all data underlying the findings in their manuscript fully available?

Reviewer #1: Yes

Reviewer #2: Yes

5. Is the manuscript presented in an intelligible fashion and written in standard English?

Reviewer #1: Yes

Reviewer #2: Yes

6. Review Comments to the Author

Reviewer #1: This manuscript was well revised about the part I have pointed out.

(Comment 3)

In page18, line2-5, the authors said that there were no statistically significant differences in clinico-pathological characteristics between patients with pM1a and pM1b and pM1c. The rate of adjuvant treatment is different between two groups.

(Response) The reviewer’s comment is correct. To clarify, we have revised as “There were no statistically significant differences in clinicopathological characteristics between patients with pM1a and pM1b and pM1c besides the rate of adjuvant treatment.” (page17, lines2-5)

→

It doesn’t make sense that “besides the rate of adjuvant treatment”.

“except for the rate of adjuvant treatment”

or

“apart from the rate of adjuvant treatment”

would be better.

Reviewer #2: (No Response)

7. PLOS authors have the option to publish the peer review history of their article (what does this mean?). If published, this will include your full peer review and any attached files.

Reviewer #1: No

Reviewer #2: No

---

## [Editor Report · Acceptance letter]

4 Nov 2020

PONE-D-20-18946R1 

Oncologic outcomes after resection of para-aortic lymph node metastasis in left-sided colon and rectal cancer 

Dear Dr. Sakamoto:

I'm pleased to inform you that your manuscript has been deemed suitable for publication in PLOS ONE. Congratulations! Your manuscript is now with our production department. 

Kind regards, 

on behalf of

Dr. Norikatsu Miyoshi 

Academic Editor

PLOS ONE